# Conjoined Genes as Common Events in Childhood Acute Lymphoblastic Leukemia

**DOI:** 10.3390/cancers14143523

**Published:** 2022-07-20

**Authors:** Marco Severgnini, Mariella D’Angiò, Silvia Bungaro, Giovanni Cazzaniga, Ingrid Cifola, Grazia Fazio

**Affiliations:** 1Istituto di Tecnologie Biomediche, Consiglio Nazionale delle Ricerche, 20054 Milano, Italy; marco.severgnini@itb.cnr.it (M.S.); ingrid.cifola@itb.cnr.it (I.C.); 2School of Medicine and Surgery, Università degli Studi di Milano Bicocca, 20126 Milano, Italy; mariella.dangio@unimib.it; 3Tettamanti Research Center, University of Milan Bicocca, 20900 Monza, Italy; g.fazio@asst-monza.it; 4Ospedale San Gerardo, Fondazione Monza e Brianza per il Bambino e la sua Mamma (MBBM), 20900 Monza, Italy; sbungaro@fondazionembbm.it

**Keywords:** childhood BCP-ALL, RNA-seq, conjoined genes, fusions

## Abstract

**Simple Summary:**

Acute lymphoblastic leukemia (ALL) is the most frequent childhood cancer. In recent years, broad application of NGS technologies enabled the discovery of novel genomically defined ALL. In this study, as a proof-of-principle, we applied RNA-seq technology to comprehensively profile the transcriptional landscape of a collection of 10 childhood BCP-ALL cases, and performed a deep bioinformatics analysis including several publicly available datasets, in order to characterize their full spectrum of transcriptional events. The paired-end RNA sequencing of our BCP-ALL pediatric cohort revealed a total of 9001 raw fusion events, which, after filtering, resulted in 245 candidate fusions. Overall, 235 out of 245 events were intra-chromosomal fusions, among which 229 involved two contiguous or overlapping genes, also known as conjoined genes (CGs). Among them, we identified a subset of 14 CGs (6.1%) exclusively expressed in leukemic cases but neither in solid cancers nor in normal samples. These events could be suggestive of a novel mechanism of transcriptional regulation in childhood leukemia and may represent novel potential leukemia-specific biomarkers.

**Abstract:**

Acute lymphoblastic leukemia (ALL) is the most frequent childhood cancer. For the last three decades, conventional cytogenetic and molecular approaches allowed the identification of genetic abnormalities having prognostic and therapeutic relevance. Although the current cure rate in pediatric B cell acute leukemia is approximately 90%, it remains one of the leading causes of mortality in childhood. Furthermore, in the contemporary protocols, chemotherapy intensity was raised to the maximal levels of tolerability, and further improvements in the outcome will depend on the characterization and reclassification of the disease, as well as on the development of new targeted drugs. The recent technological advances in genome-wide profiling techniques have allowed the exploration of the molecular heterogeneity of this disease, even though some potentially interesting biomarkers such as conjoined genes have not been deeply investigated yet. In the present study, we performed the transcriptome sequencing (RNA-seq) of 10 pediatric B cell precursor (BCP)-ALL cases with different risk (four standard- and six high-risk patients) enrolled in the Italian AIEOP-BFM ALL2000 protocol, in order to characterize the full spectrum of transcriptional events and to identify novel potential genetic mechanisms sustaining their different early response to therapy. Total RNA was extracted from primary leukemic blasts and RNA-seq was performed by Illumina technology. Bioinformatics analysis focused on fusion transcripts, originated from either inter- or intra-chromosomal structural rearrangements. Starting from a raw list of 9001 candidate events, by employing a custom-made bioinformatics pipeline, we obtained a short list of 245 candidate fusions. Among them, 10 events were compatible with chromosomal translocations. Strikingly, 235/245 events were intra-chromosomal fusions, 229 of which involved two contiguous or overlapping genes, resulting in the so-called conjoined genes (CGs). To explore the specificity of these events in leukemia, we performed an extensive bioinformatics meta-analysis and evaluated the presence of the fusions identified in our 10 BCP-ALL cohort in several other publicly available RNA-seq datasets, including leukemic, solid tumor and normal sample collections. Overall, 14/229 (6.1%) CGs were found to be exclusively expressed in leukemic cases, suggesting an association between CGs and leukemia. Moreover, CGs were found to be common events both in standard- and high-risk BCP-ALL patients and it might be suggestive of a novel potential transcriptional regulation mechanism active in leukemic cells.

## 1. Introduction

Acute lymphoblastic leukemia (ALL) is the most frequent childhood cancer. ALL onset is due to a multi-step complex process, characterized by the expansion of a pre-leukemic clone which accumulates cooperative genetic events required for full malignant transformation and clinical manifestation [1,2]. For the last three decades, several conventional cytogenetic studies of genetic aberrations that include chromosomal translocations and alterations in chromosome number have provided information on the pathogenesis of ALL. Common translocations in children with B–ALL include t(12;21) [ETV6–RUNX1](25%), t(1;19) [TCF3–PBX1] (5%), t(9;22) [BCR–ABL1] (3%) and translocations involving the MLL gene with various fusion partner genes (5%). Gains in whole chromosomes, or high hyperdiploidy (>50 chromosomes) accounts for 25% of childhood ALL, whereas hypodiploidy (< 44 chromosomes) accounts for approximately 1% of cases. Several of these genetic changes have prognostic and therapeutic implications and are important in risk stratification schemas providing more intensive and/or targeted treatment for patients at risk of developing a relapse, e.g., BCR-ABL1-positive or KMT2A rearranged, while limiting toxic effects for patients with favorable prognosis, e.g., ETV6-RUNX1-positive or hyperdiploid cases. However, despite these remarkable progresses, B cell precursor (BCP)-ALL remains one of the leading causes of mortality in childhood [3,4,5]. Furthermore, in the contemporary protocols, chemotherapy intensity was raised to the maximal levels of tolerability, and further improvements in the outcome will depend on the characterization and re-classification of cases, in particular, in the subset of BCP-ALL, in which no major genetic alteration could be detected with conventional cytogenetic and molecular approaches. Technological advances in the genomics field over the last decade, particularly with the advent of next-generation sequencing (NGS), helped unravelling ALL genomic landscape and biology. In recent years, broad application of NGS technologies, notably whole-transcriptome sequencing (RNA-seq), has redefined the molecular taxonomy of ALL. RNA-seq enabled the discovery of novel genomically defined ALL subtypes, characterized by chromosomal rearrangements cryptic on karyotyping (e.g., DUX4-rearranged ALL), new fusion genes (e.g., MEF2D, ZNF384, or NUTM1-R ALL) and expression profiles similar to classic BCP-ALL subtypes, e.g., BCR-ABL-like, ETV6-RUNX1-like [6,7]. In addition, NGS technology offered the ability to simultaneously identify heterogeneous genetic alterations, emerged as prognostically relevant such as IKZF1 deletions, sequence mutations, as well as complexly rearranged transcripts that were in general neglected or underestimated in previous reports. One such intriguing example is the “conjoined genes” (CGs), which are “read-through transcripts” or “co-transcribed genes” derived from the non-traditional splicing between two or more adjacent or overlapping genes which lie on the same chromosome. As a result, the two adjacent genes fuse together at the transcript level, with no alteration in their chromosome structure. In some cases, the transcripts formed by CGs are translated to form chimeric or completely novel proteins [8,9,10]. Therefore, they represent a new repertoire for the discovery of novel candidate biomarkers and drug targets [11].

Although CGs have been identified in several cancer types, no reports have dealt with CGs in pediatric BCP-ALL up to now [11,12,13].

In this study, we applied the RNA-seq technology as a proof-of-principle to comprehensively profile the transcriptional landscape of a collection of childhood BCP-ALL cases with different recurrence risk, and performed a deep bioinformatics analysis including also several publicly available datasets, in order to characterize their full spectrum of transcriptional events and to reveal the comprehensive expression of CGs.

## 2. Materials and Methods

Ten pediatric BCP-ALL cases were profiled by whole-transcriptome RNA-seq technology. All the cases selected were already enrolled in the Italian AIEOP-BFM ALL2000 clinical protocol. They were homogeneous for all clinical or genetic risk factors but differed by minimal residual disease (MRD) after induction (four standard-(SR) and six high-risk (HR) patients, according to MRD at day 33 and 78 of treatment) (Table 1).

Total RNA was extracted from primary bone marrow (BM) leukemic blasts using the guanidine thiocyanate-phenol-chloroform method and checked for integrity by microcapillary electrophoresis on the 2100 Bioanalyzer instrument (Agilent Technologies, Santa Clara, CA, USA). Starting from 2–3 μg of total RNA per sample, poly-A+ RNA-seq libraries were prepared using the TruSeq RNA Sample Prep Kit (Illumina, San Diego, CA, USA), according to manufacturer’s instructions, and sequenced on the Genome Analyzer IIx platform (Illumina) in 76-cycle paired-end runs, generating a mean of 80 M raw reads/sample.

After fastq quality control by using FastQC tool (https://www.bioinformatics.babraham.ac.uk/projects/fastqc/, accessed on 15 April 2022), candidate fusions were searched on raw sequencing reads by FusionMap (v.10.0, 9) [14], using human HG38 as the reference genome and Gencode v28 (excluding known read-through transcripts) as the gene model. The “FilterUnlikelyFusionReads” parameter was set to „False” in order to retrieve also read-through events. Then, we implemented a custom bioinformatics pipeline to identify all the putative fusions, to filter them according to stringent qualitative criteria and remove false positives, and, thus, to identify a subset of confident fusion events (“candidate fusions”) originated from chromosomal rearrangements (inter- or intra-chromosomal translocations) or not (conjoined genes) (Figure 1). Briefly, the raw list of events was filtered out excluding those present in FusionMap own blacklist, those with the same chromosomal breakpoint, those having a complete match of the candidate junction on the genome, those with a read depth <10 in the region flanking the rearrangement or an incidence of the reads supporting the event <5% of the total and those with no reads spanning across the junction after remapping the original reads on the FusionMap-generated transcript sequence.

Candidate fusions of interest were validated in the original samples by RT-PCR and/or FISH assays. SNP array copy number profiles already produced for these samples on Cytogenetics Whole Genome 2.7 M Arrays (Affymetrix, Santa Clara, CA, USA) were exploited to assess the presence of chromosomal imbalances accompanying fusion events.

To explore the specificity of these events in leukemia, a representative selection of these candidate fusion transcripts was experimentally evaluated in a commercial RNA library of human normal tissues derived from healthy donors (Human Total RNA Master Panel II, Diatech LabLine, Jesi, Italy), including lung, trachea, skeletal muscle, brain, prostate, testis, uterus, adrenal gland, spleen, thymus, salivary gland, stomach, thyroid and kidney tissues. Moreover, we performed an extensive bioinformatics meta-analysis to evaluate the incidence of these candidate fusions in other publicly available RNA-seq datasets, including: 1 AML study (27 cases from Leucegene project); 2 T-ALL studies (12 cases from Leucegene project, and 14 cases from COG study); 1 B-ALL study (10 cases at diagnosis and 10 cases at relapse); 10 solid cancer types from TCGA project (bladder urothelial carcinoma (BLCA), breast carcinoma (BRCA), cervical squamous cell carcinoma (CESC), colon adenocarcinoma (COAD), kidney renal clear cell carcinoma (KIRC), low grade glioma (LGG), lung adenocarcinoma (LUAD), prostate adenocarcinoma (PRAD), skin cutaneous melanoma (SKCM), thyroid carcinoma (THCA), 20 cases for each cancer type, for a total of 200 cases); 1 CEU population dataset from the 1000 genomes sample collection (91 samples from Geuvadis consortium). All the downloaded cases are listed in Appendix A. Fastq files were checked by FastQC tool and then mapped on the junctions of the candidate fusions, in order to find reads spanning over them. A specific filter was implemented with the aim of excluding false positive matches with reads mapping only on one side of the junction. Finally, events were screened against the FusionHub [15] and the Atlas of Genetics Oncology [16] databases, in order to filter out candidates already described as associated to non-tumoral samples, and annotated by searching support in already known gene models (Gencode, GenBank, RefSeq, UCSC genes and VEGA).

## 3. Results

The paired-end RNA sequencing of our BCP-ALL pediatric cohort revealed a total of 9001 raw fusion events, which, after filtering, resulted in 245 candidate fusions. On average, over the 10 BCP-ALL cases, 52% of the raw events were filtered out because they were in the FusionMap blacklist, 19% involved the same breakpoints over multiple events, 38% had a complete match of the candidate junction on the genome, 22% had <10× coverage, 37% were characterized by having <5% incidence over the linear transcript and 8% were excluded since no support was found when remapping the original sequencing read on the junction. Overall, 235 out of 245 events were intra-chromosomal fusions. Among them, 229 involved two contiguous or overlapping genes (CGs), with 221 (97%) CGs generated by genes in the same orientation, while 8 (3%) by genes in opposite direction. 204 CGs (89%) were identified both in cancer patients and in normal tissues, whereas 11 were detected exclusively in cancer, both in leukemia and solid tumors (Table 2).

Interestingly, the other 14/229 (6.1%) CGs were found to be exclusively expressed in leukemic cases but neither in solid cancers nor in normal samples (Table 3). Three of them *(KLHL22::SCARF2, PPP1R3F::LL0XNC01-7P3.1*, and *FAM200B::BST1*) were supported by known gene models.

In particular, *KLHL22::SCARF2* was found as a transcript variant (ENST00000429594) of *SCARF2* gene, annotated as “nonsense-mediated decay”, and was present also in the ConjoinG database (CGHSA0597); at the same time, *PPP1R3F::LL0XNC01-7P3.1* was previously reported in GenBank (LF211393) as a “Polycomb-Associated Non-Coding RNA” [17]. Some CGs identified only in leukemia samples involved long non-coding RNAs. Furthermore, by transcriptomic analysis we were able to identify 10 fusions compatible with inter-chromosomal translocations not previously showed by conventional methods (*PAX5::POM121C, IK::FBXW2, ZNF444::HLA-B, NFX1::DICER1, DCAF8::ZNF836, DMD::STAMBPL1, SLFNL1::SMPD2, RP11-148O21.2::ATG4B, TMEM263::CD47,INPP5A::SETD7*) and 6 intra-chromosomal events involving distant genes (*NUP214::ABL1, MAEA::CTBP1, ZC3H12D::RP11-445F6.2, MAML2::FAT3, MNT::CLUH, TSKS::ARRDC2*) (Table 4). Details about reads supporting both CGs and fusions are depicted in Appendix A.

To explore the specificity of these events, a representative selection of CGs and translocations was evaluated by RT-PCR, SNP arrays and Sanger sequencing (Figure 2A–C).

The novel *PAX5::POM121C* fusion was identified in one SR BII ALL patient, confirmed by RT-PCR and Sanger sequencing. Additionally, *NUP214::ABL1* fusion was identified in only one HR BI ALL case and confirmed by the same methods. Most of the remaining rearrangements were not experimentally investigated; however, they were identified in silico in other public RNA-seq datasets, which can be considered as an indirect validation. Considering the total of 16 fusion transcripts and 25 cancer-specific CGs, all our BCP-ALL patients, except for one, carried more than one transcriptional rearrangement; the mean number of events was 4.5 for each patient, with range from 1 to 9, demonstrating the complex transcriptional landscape of leukemia. According to MRD classification, the four SR patients presented a mean number of transcriptional rearrangements of 5 (range 3–7), while in the HR group, the number was 4 but with a wider range (1–9 events). No evident differences were observed both in the entire cohort and in the two MRD groups, probably due to the low number of cases.

## 4. Discussion

Pediatric cancers, even leukemia, differ from adult tumors, especially for their very low mutational rate.

Therefore, their etiology may involve further oncogenic mechanisms, such as the development of chimeric transcripts [18,19]. Here, by applying the RNA-seq technology, we comprehensively profiled the transcriptional landscape of 10 childhood BCP-ALL patients negative for recurrent translocations and with different recurrence risk. Overall, we identified 16 different transcriptional fusions not previously identified by conventional cytogenetic, comprising 10 translocations and 6 intra-chromosomal events compatible with deletions. In particular, the RNA-seq technology allowed us to identify a new *PAX* gene translocation (*PAX5::POM121C,* as we already reported in [20]), and a fusion involving *NUP214* and *ABL1* genes, respectively, in one SR and one HR B-ALL case. *NUP214::ABL1* was originally reported as a recurrent abnormality in T-ALL, accounting for 6% of adult and less than 2% of pediatric T-ALL, as recently reported by the Associazione Italiana di Onco-Ematologia Pediatrica [21]. Otherwise, only few cases have been reported in B-ALL [22,23,24]. Moreover, the fusion involving *MAEA::CTBP1* (detected in one SR B-ALL case from our cohort), although with different breakpoints, has already been described in colon adenocarcinoma and acute myeloid leukemia [25].

Besides chromosomal rearrangements, RNA processing events, such as cis- and trans-splicing, also contribute to the formation of chimeric RNAs. Alternative splicing between exons of neighboring genes is a RNA processing event that occurs within a single pre-mRNA, where the transcription machinery reads through the intergenic regions of the two genes. Although only few examples of spliced RNA chimeras were experimentally confirmed in mammalian cells, bioinformatics analysis of paired-end RNA-seq data have successfully identified many chimeric RNAs composed of two adjacent genes, which could originate from transcriptional read-through [11]. Since it is evident CGs are not merely artifacts of transcription, then they must be the result of some specific genomic requirements and have well-defined functional roles. This idea is further strengthened by the fact that some CGs have endured purifying evolutionary selective pressure and are conserved in different animal species. In addition to protein evolution, CGs can be responsible for gene regulation by preventing the expression of at least one or more of the parent genes [10].

Until recently, the assumption was that all gene fusions and fusion products (RNAs and proteins) were exclusive to cancer. This dogma has been challenged as more groups demonstrated the presence of fusion RNAs and proteins in non-pathological situations. Thus, many CGs have been discovered in both normal and cancer cells and collected in different databases, such as the ConjoinG database, dedicated to the 800 CGs identified in the human genome up to now (https://metasystems.riken.jp/conjoing/, accessed on 15 April 2022) [26,27,28,29,30]. The implications of these intergenic spliced chimeric RNAs are multifaceted. On one hand, their presence in normal tissues and cells makes doubtful the use of fusion RNAs in cancer diagnosis and treatment. On the other hand, even though they represent new biomarkers and drug targets, no reports have dealt with CGs in pediatric BCP-ALL.

In our cohort, 229 out of the 245 fusion events identified by RNA-seq were CGs involving two contiguous or overlapping genes. As previously reported in literature, most of them (89%) were expressed also in normal samples, so that they were not specific for leukemia. Nonetheless, we identified a subset of 14 CGs (6.1%) exclusively expressed in leukemic cases but neither in solid cancers nor in normal samples. These events could be suggestive of a novel mechanism of transcriptional regulation in childhood leukemia and may represent novel potential leukemia-specific biomarkers. Some of them, even when involving cancer genes, were only indirectly validated by identification in other public RNA-seq datasets. This could be related to the higher sensitivity of RNA-seq technology as compared to other molecular assays (RT-PCR, FISH and SNP array) to identify sub-clonal genetic events and confirm leukemia as an oligoclonal disease. Some of the leukemia-specific CGs identified in our cohort involved long non-coding RNAs. Although discoveries on long non-coding RNAs are increasing, their function both in normal and malignant tissue remains unclear [31]. As previously reported in other hematological malignancies, non-coding RNAs involved in chimeric transcripts might lead to dysregulation of fusion partner gene expression [32,33]. In particular, in B-other ALL, this genetic mechanism could be of interest to explain some gene expression alterations in leukemic cells. We also observed that CGs were present and recurrent both in MRD stratified standard- and high-risk patients, suggesting a possible role on the disease onset rather than on treatment response. However, due to the small size of the cohort analyzed, no conclusion can be reached and certainly an extensive leukemia cohort is needed to define the possible role of CGs on chemotherapy response or risk stratification at diagnosis. Most of our cases carried more than one transcriptional event. This observation corroborates the complex and heterogeneous transcriptional landscape of acute leukemia, in particular for those patients not carrying recurrent translocations, as in our cohort. Moreover, the co-presence of more CGs in the same leukemia case suggests their possible role as cooperative rather than primary genetic events in leukemogenesis.

## 5. Conclusions

In conclusion, even in a very small cohort of children we demonstrated the presence of CGs specific for acute lymphoblastic leukemia, most of them involving non-coding RNAs. This study is a proof of principle for further studies crucial to confirm and to expand the knowledge on the possible role of CGs in children and adult ALL. In particular, the evaluation of CGs in patients without any recurrent translocation at diagnosis in an extensive cohort of adult and pediatric patients might be of interest to dissect the role of these rearrangements in leukemogenesis and risk assessment.

In this setting, the extensive use of NGS approaches at ALL diagnosis, in particular whole transcriptome analysis, could be a useful tool to further dissect and identify potential novel mechanisms of transcriptional regulation in leukemic cells, in order to clarify the relevant genetic mechanisms leading to the development of the disease.

## Figures and Tables

**Figure 1 cancers-14-03523-f001:**
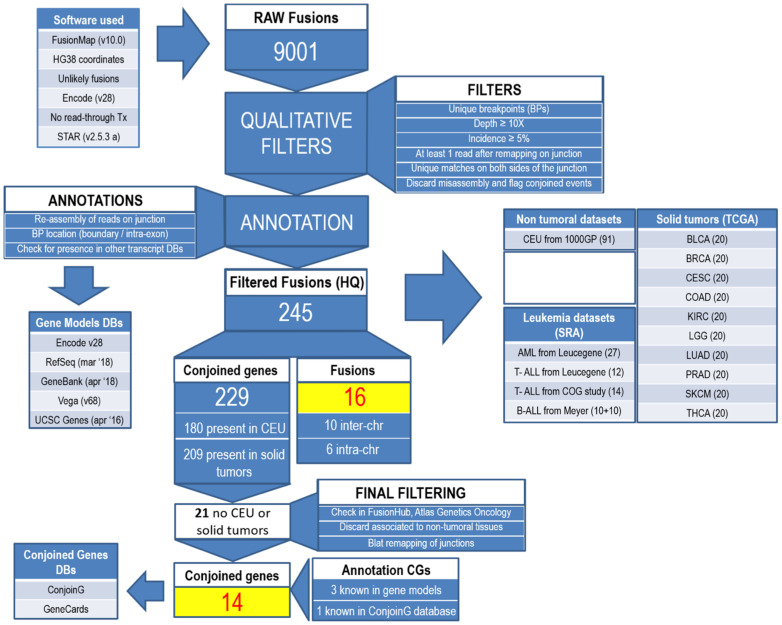
Bioinformatics pipeline.

**Figure 2 cancers-14-03523-f002:**
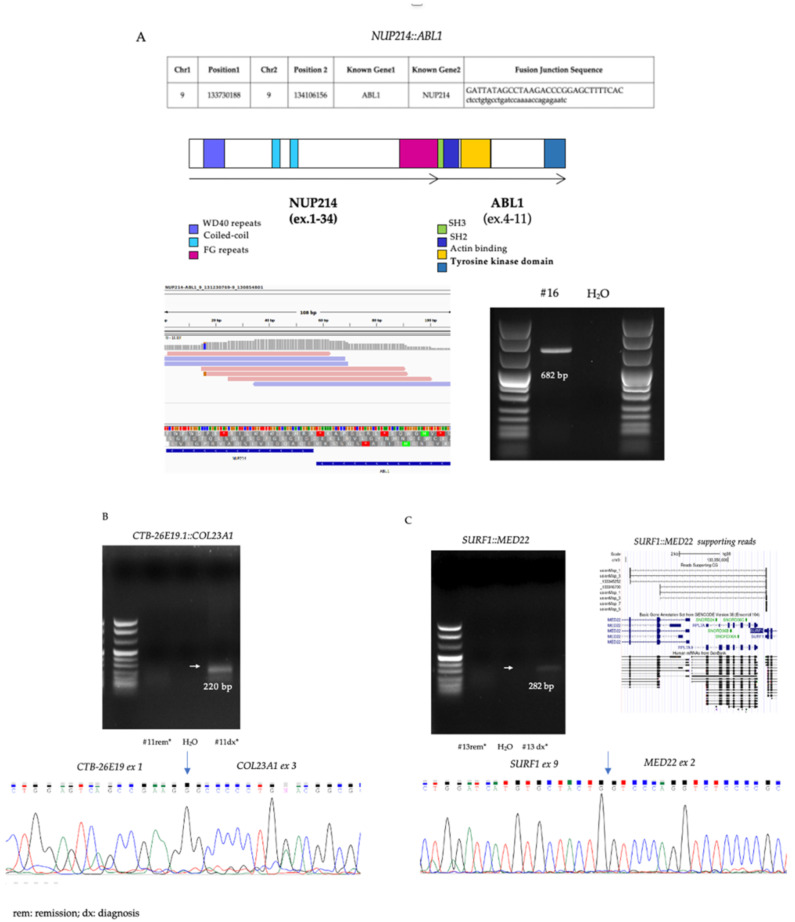
(**A**) RT-PCR validation of *NUP214::ABL1* fusion. (**B**) RT-PCR and Sanger sequencing validation of *CTB-26E19.1::COL23A1* conjoined gene. (**C**) RT-PCR, Sanger sequencing and supporting reads validation of *SURF1::MED22* conjoined gene.

**Table 1 cancers-14-03523-t001:** Patients‘ characteristics.

Code	Age	Sex	Phenotype	BM Blasts %	Translocations	DNA Index	Steroid Response	MRD Risk Stratification
#8	16	M	BII	86	NEG	1	PGR	HR
#11	15	M	BII	80	NEG	1	PGR	HR
#12	3	F	BI	73	NEG	1	PGR	HR
#13	4	F	BIII	78	NEG	1	PGR	HR
#15	9	F	BII	74	NEG	1	PGR	HR
#16	11	M	BIII	90	NEG	1	PGR	HR
#6	1	M	BII	94	NEG	1	PGR	SR
#7	5	M	BII	70	NEG	1	PGR	SR
#9	15	M	BII	94	NEG	1	PGR	SR
#14	8	M	BII/BIII	90	NEG	1	PGR	SR

Translocations: t(9;22), t(4;11), t(12;21), t(1;19); PGR: prednisone good response after 7 days of steroid treatment; HR: high risk; SR: standard risk.

**Table 2 cancers-14-03523-t002:** CGs shared by leukemia and solid tumors.

Conjoined Genes	Gene Orientation *	5′ Gene	3′ Gene	5′ Breakpoint	3′ Breakpoint	Annotation	Our ALL Cohort (Case ID#)	AML **	T-ALL (Leucegene) **	T-ALL (COG) **	B-ALL**	TCGA Solid Tumors ***
LRRC61::ZBED6CL	+/+	LRRC61	ZBED6CL	7:150328742	7:150330107	-	1/10 (#14)	17/27	8/12	6/14	0/20	THCA (1/20)
ZNF250::COMMD5	+/+	ZNF250	COMMD5	8:144880365	8:144851395	GenBank	1/10 (#14)	15/27	9/12	4/14	4/20	LGG (1/20), PRAD (1/20)
RP11-305L7.3::RP11-305L7.1	+/+	RP11-305L7.3	RP11-305L7.1	9:91163005	9:91106671	RefSeq	1/10 (#13)	10/27	8/12	3/14	1/20	BLCA (1/20), BRCA (1/20), SKCM (1/20)
RP11-556E13.1::RP11-346D6.6	+/+	RP11-556E13.1	RP11-346D6.6	10:52561006	10:52463332		1/10 (#11)	9/27	2/12	0/14	0/20	LUAD (1/20), SKCM (1/20)
OPN4::LDB3	+/+	OPN4	LDB3	10:86666848	10:86668669	GenBank, RefSeq	1/10 (#16)	0/27	0/12	0/14	2/20	BRCA (1/20), CESC (1/20), KIRC (6/20), LGG (9/20), SKCM (1/20)
C14orf37::RP11-999E24.3	+/+	C14orf37	RP11-999E24.3	14:58003907	14:57994417	RefSeq	1/10 (#13)	2/27	0/12	0/14	0/20	KIRC (1/20), LGG (1/20)
RP11-1360M22.11::RP11-810K23.8	+/+	RP11-1360M22.11	RP11-810K23.8	15:20012876	15:21019524	-	1/10 (#13)	5/27	3/12	4/14	0/20	KIRC (1/20), LGG (1/20)
IGF1R::RP11-35O15.2	+/+	IGF1R	RP11-35O15.2	15:98649675	15:98660230	GenBank	2/10 (#6,#7)	8/27	3/12	0/14	3/20	BRCA (1/20), CESC (2/20), LGG (1/20)
NHLRC4::PIGQ	+/+	NHLRC4	PIGQ	16:568989	16:574066	Ensembl, UCSC, Vega, ConjoinG, FusionHub	2/10 (#7,#8)	12/27	7/12	6/14	10/20	BRCA (2/20), COAD (2/20), KIRC (1/20)
MAPK3::GDPD3	+/+	MAPK3	GDPD3	16:30114203	16:30113437	-	1/10 (#11)	10/27	5/12	4/14	4/20	BRCA (1/20), LGG (1/20)
TRAPPC1::KCNAB3	+/+	TRAPPC1	KCNAB3	17:7930527	17:7927826	ConjoinG, FusionHub	1/10 (#11)	0/27	1/12	2/14	0/20	CESC (2/20)

* +/+: Genes with same orientation. ** Frequency for each cancer dataset tested in this study. *** THCA: thyroid carcinoma; LGG: low grade glioma; PRAD: prostate adenocarcinoma; BLCA: bladder urothelial carcinoma; BRCA: breast invasive carcinoma; SKCM: skin cutaneous melanoma; LUAD: lung adenocarcinoma; CESC: cervical squamous cell carcinoma and endocervical adenocarcinoma; KIRC: kidney renal clear cell carcinoma; COAD: Colon adenocarcinoma.

**Table 3 cancers-14-03523-t003:** Leukemia-specific conjoined genes.

Conjoined Genes	Gene Orientation *	5′ Gene	3′ Gene	5′ Breakpoint	3′ Breakpoint	Annotation	Our ALL Cohort (Case Id#)	AML **	T-ALL (Leucegene) **	T-ALL (COG) **	B-ALL **
CTB-26E19.1::COL23A1	+/+	CTB-26E19.1	COL23A1	5:178442931	5:178306919	-	1/10 (#11)	2/27	7/12	7/14	3/20
SURF1::MED22	+/+	SURF1	MED22	9:133351836	9:133345252	-	2/10 (#6,#7)	1/27	2/12	0/14	0/20
SURF1::MED22	+/+	SURF1	MED22	9:133351836	9:133346700	-	1/10 (#13)	6/27	4/12	0/14	1/20
CACUL1::RP11-427L15.2	+/+	CACUL1	RP11-427L15.2	10:118693702	10:118692438	-	2/10 (#7,#11)	3/27	1/12	0/14	1/20
TMEM86A::RP11-1081L13.4	+/+	TMEM86A	RP11-1081L13.4	11:18702116	11:18706818	-	1/10 (#7)	1/27	1/12	1/14	0/20
RP11-20D14.3::RIMKLB	+/+	RP11-20D14.3	RIMKLB	12:8668831	12:8713811	-	9/10 (#6,#7,#9,#8,#11,#12,#13,#15,#16)	17/27	1/12	1/14	16/20
RP11-397H6.1::RP11-541G9.1	+/+	RP11-397H6.1	RP11-541G9.1	12:97033423	12:97185076	-	1/10 (#13)	0/27	0/12	0/14	0/20
CCPG1::PIGBOS1	+/+	CCPG1	PIGBOS1	15:55355634	15:55317867	-	2/10 (#6,#13)	14/27	2/12	6/14	0/20
AC019118.3::AC019118.2	+/+	AC019118.3	AC019118.2	2:3145547	2:2966675	-	2/10 (#14,#16)	0/27	1/12	0/14	4/20
RP11-87G24.3::RP11-87G24.6	+/+	RP11-87G24.3	RP11-87G24.6	17:76963893	17:76957657	-	6/10 (#7,#8,#12,#13,#14,#16)	3/27	9/12	6/14	6/20
KLHL22::SCARF2	+/+	KLHL22	SCARF2	22:20446443	22:20430557	Ensembl, UCSC, Vega, ConjoinG, FusionHub	4/10 (#9,#12,#13,#15)	14/27	8/12	5/14	4/20
PPP1R3F::LL0XNC01-7P3.1	+/+	PPP1R3F	LL0XNC01-7P3.1	X:49270873	X:49273514	GenBank	1/10 (#7)	2/27	0/12	0/14	0/20
MYNN::RP11-362K14.7	+/+	MYNN	RP11-362K14.7	3:169786730	3:169793627	-	2/10 (#8,#14)	8/27	4/12	4/14	0/20
FAM200B::BST1	+/+	FAM200B	BST1	4:15687148	4:15711807	RefSeq	2/10 (#13,#14)	3/27	2/12	0/14	0/20

* +/+: Genes with same orientation. ** Frequency for each leukemia dataset tested in this study.

**Table 4 cancers-14-03523-t004:** Fusions identified in our ALL cohort.

Fusions	Gene Orientation *	5′ Gene	3′ Gene	5′ Breakpoint	3′ Breakpoint	Annotation	Our ALL Cohort (Case ID#)	AML **	T-ALL (Leucegene) **	T-ALL (COG) **	B-ALL **
IK::FBXW2	+/−	IK	FBXW2	5:140659164	9:120792947	-	1/10 (#13)	0/27	0/12	0/14	0/20
ZNF444::HLA-B	+/−	ZNF444	HLA-B	19:56160570	6:31355372	-	1/10 (#9)	1/27	3/12	0/14	0/20
PAX5::POM121C	+/+	PAX5	POM121C	9:36966549	7:75441583	-	1/10 (#6)	0/27	0/12	0/14	0/20
NFX1::DICER1	+/−	NFX1	DICER1	9:33290597	14:95141748	-	1/10 (#9)	0/27	0/12	0/14	0/20
DCAF8::ZNF836	−/−	DCAF8	ZNF836	1:160220067	19:52156511	-	1/10 (#14)	0/27	0/12	0/14	0/20
DMD::STAMBPL1	−/+	DMD	STAMBPL1	X:32216916	10:88893871	-	1/10 (#13)	0/27	0/12	0/14	0/20
SLFNL1::SMPD2	−/+	SLFNL1	SMPD2	1:41022123	6:109441113	-	1/10 (#16)	0/27	0/12	0/14	0/20
RP11-148O21.2::ATG4B	−/+	RP11-148O21.2	ATG4B	8:11558702	2:241672559	-	1/10 (#7)	0/27	0/12	0/14	0/20
TMEM263::CD47	+/−	TMEM263	CD47	12:106971137	3:108055566	-	1/10 (#11)	0/27	0/12	0/14	0/20
INPP5A::SETD7	+/−	INPP5A	SETD7	10:132607956	4:139548119	-	1/10 (#7)	0/27	0/12	0/14	0/20
ZC3H12D::RP11-445F6.2	−/+	ZC3H12D	RP11-445F6.2	6:149456666	6:139271659	-	1/10 (#14)	0/27	0/12	0/14	0/20
NUP214::ABL1	−/−	NUP214	ABL1	9:131230769	9:130854801	FusionHub, Atlas Genetics Oncology	1/10 (#16)	0/27	0/12	0/14	0/20
MAML2::FAT3	−/+	MAML2	FAT3	11:96091892	11:92352096	-	1/10 (#15)	0/27	0/12	0/14	0/20
MNT::CLUH	−/−	MNT	CLUH	17:2400640	17:2704564	-	2/10 (#9,#11)	3/27	2/12	0/14	0/20
TSKS::ARRDC2	−/+	TSKS	ARRDC2	19:49746470	19:18007338	-	1/10 (#11)	0/27	0/12	0/14	0/20
MAEA::CTBP1	+/−	MAEA	CTBP1	4:1289982	4:1238337	FusionHub, Atlas Genetics Oncology	2/10 (#7,#12)	9/27	3/12	6/14	2/20

* +/+, −/−: Genes with same orientation; +/−, −/+: Genes with opposite orientation. ** Frequency for each leukemic dataset tested in this study.

## Data Availability

Data supporting reported in Appendix A.

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
