# Peer review of "Conjoined Genes as Common Events in Childhood Acute Lymphoblastic Leukemia"

_cancers, 2022, doi:10.3390/cancers14143523_

Round 1

Reviewer 1 Report

The authors collected 10 childhood BCP-ALL samples, and all samples were performed with RNA-seq, resulting in 229 CGs. In total, three CGs were verified by wet-lab methods; one of which was verified by FISH in a published paper; two of which were performed with Sanger sequencing in this manuscript. I tried to confirm these two Sanger sequencing results by blasting the two sequences the authors provided in Fig 2b and 2c, but unfortunately, neither sequence can be blasted with the right genes.

Overall, most of the results were low-quality and not verified, and I was unable to find any confirmed results from the manuscript, so I don't think this paper can be published.

This manuscript is a resubmission of an earlier submission. The following is a list of the peer review reports and author responses from that submission.

Round 1

Reviewer 1 Report

In this manuscript, Marco and colleagues report their results on RNA sequencing analysis of n=10 diagnostic specimens obtained from pediatric patient suffering from acute lymphoblastic leukemia (ALL), the most frequent malignant disease in children and adolescents, enrolled in the AIEOP-BFM 2000 protocol. All cases employed into this study were negative for recurrent gene fusions used for therapy stratification and comprised sample from both, standard and high-risk stratified patients according to molecular MRD measurements. Interestingly, the authors report on detection of a majority of gene fusions deriving from intrachromosomal fusions and again with the majority being form overlapping or adjacent genes. They report a subset of these conjoined genes (CGs) to be described for other cancers and, interestingly, identify 14 CG to be exclusively expressed in ALL. Moreover, the authors also describe 10 new inter-chromosomal fusions. 

This is an interesting and well written report, however I would have the following remarks:

Major points:

1. Data presentation: In addition to the CGs already reported in other cancers, the CGs as well as the newly described inter-chromosomal fusions should be presented in tables. For the CGs: which were identified in which samples? Overlap and distribution of identified CGs within different samples?

2. Patient characteristic are missing, please provide, including findings in conventional genetic diagnostics.

3. Validation: Results on the validation of the identified (selected) fusions need to be included into the ms. This information is of crucial importance for the findings reported, ‘data not shown’ is not acceptable in this case. 

validation of (selected) fusions in an additional/extended sample cohort would clearly further strengthen the findings and improve the ms.

4. Potential function and role of the identified fusions: This is obviously of great importance for the study, however not easily addressed. However, more information of potential mechanisms/discussion of possible impact of the genes involved for selected fusions/examples) would be of interest, in particular in the light of the speculation on using these fusions as indicators for treatment response or outcome. 

Author Response

Comments and Suggestions for Authors

In this manuscript, Marco and colleagues report their results on RNA sequencing analysis of n=10 diagnostic specimens obtained from pediatric patient suffering from acute lymphoblastic leukemia (ALL), the most frequent malignant disease in children and adolescents, enrolled in the AIEOP-BFM 2000 protocol. All cases employed into this study were negative for recurrent gene fusions used for therapy stratification and comprised sample from both, standard and high-risk stratified patients according to molecular MRD measurements. Interestingly, the authors report on detection of a majority of gene fusions deriving from intrachromosomal fusions and again with the majority being form overlapping or adjacent genes. They report a subset of these conjoined genes (CGs) to be described for other cancers and, interestingly, identify 14 CG to be exclusively expressed in ALL. Moreover, the authors also describe 10 new inter-chromosomal fusions. 

This is an interesting and well written report, however I would have the following remarks:

Major points:

  1. Data presentation: In addition to the CGs already reported in other cancers, the CGs as well as the newly described inter-chromosomal fusions should be presented in tables. For the CGs: which were identified in which samples? Overlap and distribution of identified CGs within different samples?

Reply: According to Reviewer’s suggestion, we presented the 14 leukemia-specific CGs and the 16 newly described fusions in the new Tables 3 and 4, respectively.

  1. Patient characteristic are missing, please provide, including findings in conventional genetic diagnostics.

 Reply: According to Reviewer’s suggestion, we presented the patients’ characteristics in table 1.

  1. Validation: Results on the validation of the identified (selected) fusions need to be included into the ms. This information is of crucial importance for the findings reported, ‘data not shown’ is not acceptable in this case. 

validation of (selected) fusions in an additional/extended sample cohort would clearly further strengthen the findings and improve the ms.

Reply: According to Reviewer’s suggestion, we include in the tables 2, 3 and 4 information about validation and we added figure 2A, B  and C with some examples of RT-PCR validation in conjoined and fusion transcripts. Moreover reads supporting CGs and fusions have been shown in supplementary figures S2.

  1. Potential function and role of the identified fusions: This is obviously of great importance for the study, however not easily addressed. However, more information of potential mechanisms/discussion of possible impact of the genes involved for selected fusions/examples) would be of interest, in particular in the light of the speculation on using these fusions as indicators for treatment response or outcome.

Reply: According to Reviewer’s suggestion, we integrated in the discussion the potecial role of CGs in agreement with scientific literature.

Reviewer 2 Report

Marco et al. applied RNA-seq technology to analyze the conjoined genes (CGs) in 10 childhood Acute Lymphoblastic Leukemia (ALL) cases. Among the identified CGs from these cases, they identified a total of 14 CGs exclusively expressed in leukemic cases but not in solid cancers nor in normal samples. However, the methods and results were not adequately presented in the manuscript. The reviewer has the following concerns:

Major:

1.       1. Please provide the demographic and clinical information for all 10 ALL cases.

2.       2. Table 1 is too simple, at least some columns, including 3’ gene, 3’ gene breakpoint, 5’ gene, 5’ gene breakpoint, frequency in the ALL database, frequency in the XXX (cancer type) database, validation methods (RT-PCR, Sanger sequencing, FISH, etc.), should be included in Table 1.

3.      3. The authors should also provide Tables to display the main information I listed in comment 2 for the interested fusions/GCs (14 unique ALL GCs, 10 inter-chromosomal fusions, 6 intra-chromosomal fusions, etc.)

4.       4. [Page 3] “Candidate fusions of interest were validated in the original samples by RT-PCR and/or FISH assays. SNP array copy number profiles already produced for these samples …”:     For each validated fusion, the authors should plot a figure to display the RNA spanning reads supporting the fusion (example: Figure 1 in PMID: 29929942, DOI: 10.1016/j.jmoldx.2018.03.007) and images/figures to show the experimental validation results.

5.       5. For the 14 CGs that were found to be exclusively expressed in leukemic cases but neither in solid cancers nor in normal samples, the authors should plot a figure to show the read coverage across genes for each fusion (example: Figure 1B in PMID: 32466770, DOI: 10.1186/s13059-020-02043-x).

6.       6. [Page 4] “Interestingly, the other 14/229 (6.1%) CGswere found to be exclusively expressed in leukemic cases but neither in solid cancers nor in normal samples (CTB-26E19.1::COL23A1, SURF1::MED22, CACUL1::RP11-427L15.2, TMEM86A::RP11-1081L13.4, RP11-20D14.3::RIMKLB, RP11-397H6.1::RP11-541G9.1, CCPG1::PIGBOS1, AC019118.3::AC019118.2, RP11-87G24.3::RP11-87G24.6, KLHL22::SCARF2, PPP1R3F::LL0XNC01-7P3.1, MYNN::RP11-362K14.7, FAM200B::BST1).”: The author mentioned 14 CGs, but only listed 13.

7.       7. The Introduction and Abstract are so rough, the authors should, at least summarize the necessity of fusion analysis in ALL therapies. They also didn’t summarize the previous fusion studies for ALL and their shortcomings.

8.      8. The title is ambiguous. I don’t know whether the authors aimed to emphasize that conjoined genes are common in childhood ALL or conjoined genes usually occur many times in one childhood ALL case.

Minor:

1.      1. Typo: “Interestingly, the other 14/229 (6.1%) CGswere found to be exclusively expressed in leukemic cases but neither in solid cancers nor in normal samples …”

2.       2. [Abstract] “Among them, we identified a subset of 14 CGs (6.1%) exclusively expressed in leukemic cases but not in solid cancers neither in normal samples.”:     “neither” --> “nor”

Author Response

Comments and Suggestions for Authors

Marco et al. applied RNA-seq technology to analyze the conjoined genes (CGs) in 10 childhood Acute Lymphoblastic Leukemia (ALL) cases. Among the identified CGs from these cases, they identified a total of 14 CGs exclusively expressed in leukemic cases but not in solid cancers nor in normal samples. However, the methods and results were not adequately presented in the manuscript. The reviewer has the following concerns:

 Major:

  1. Please provide the demographic and clinical information for all 10 ALL cases.

 Reply: According to Reviewer’s suggestion, we presented the patients’ characteristics in table 1.

  1.     Table 1 is too simple, at least some columns, including 3’ gene, 3’ gene breakpoint, 5’ gene, 5’ gene breakpoint, frequency in the ALL database, frequency in the XXX (cancer type) database, validation methods (RT-PCR, Sanger sequencing, FISH, etc.), should be included in Table 1.

Reply: According to Reviewer’s suggestion, we included a new version of Table 1 (now table 2), providing details about the listed 11 CGs, such as 5’ and 3’ genes, breakpoint positions, frequency in our ALL cohort as well as in the other datasets downloaded for leukemias and solid tumors.

  1.     The authors should also provide Tables to display the main information I listed in comment 2 for the interested fusions/GCs (14 unique ALL GCs, 10 inter-chromosomal fusions, 6 intra-chromosomal fusions, etc.)

Reply: As requested, we provided information for the 14 leukemia-specific CGs and for the 16 fusions in the new Tables 3 and 4, respectively.

  1.      [Page 3] “Candidate fusions of interest were validated in the original samples by RT-PCR and/or FISH assays. SNP array copy number profiles already produced for these samples …”: For each validated fusion, the authors should plot a figure to display the RNA spanning reads supporting the fusion (example: Figure 1 in PMID: 29929942, DOI: 10.1016/j.jmoldx.2018.03.007) and images/figures to show the experimental validation results.

Reply: According to Reviewer’s suggestion, we include in the table information about validation and we added figure 2 A, B and C with some example of RT-PCR validation in conjoined and fusion transcripts.

  1.      For the 14 CGs that were found to be exclusively expressed in leukemic cases but neither in solid cancers nor in normal samples, the authors should plot a figure to show the read coverage across genes for each fusion (example: Figure 1B in PMID: 32466770, DOI: 10.1186/s13059-020-02043-x).

Reply: According to Reviewer’s suggestion, we include in the table information about validation and we added figure 2 A, B  and C with some examples of RT-PCR validation in conjoined and fusion transcripts. Moreover reads supporting CGs and fusions have been shown in supplementary figures S2.

  1.      [Page 4] “Interestingly, the other 14/229 (6.1%) CGswere found to be exclusively expressed in leukemic cases but neither in solid cancers nor in normal samples (CTB-26E19.1::COL23A1, SURF1::MED22, CACUL1::RP11-427L15.2, TMEM86A::RP11-1081L13.4, RP11-20D14.3::RIMKLB, RP11-397H6.1::RP11-541G9.1, CCPG1::PIGBOS1, AC019118.3::AC019118.2, RP11-87G24.3::RP11-87G24.6, KLHL22::SCARF2, PPP1R3F::LL0XNC01-7P3.1, MYNN::RP11-362K14.7, FAM200B::BST1).”: The author mentioned 14 CGs, but only listed 13.

Reply: We thank the reviewer for this comment. The CGs here identified are effectively 14, we listed 13 events since we found two CGs involving the same SURF1::MED22 but with different breakpoints. To avoid such misunderstanding, we showed the 14 leukemia-specific CGs in the new Table 2, and removed the list from the text of the manuscript.

  1.      The Introduction and Abstract are so rough, the authors should, at least summarize the necessity of fusion analysis in ALL therapies. They also didn’t summarize the previous fusion studies for ALL and their shortcomings.

Reply: According to the rewiever comment we implemented the abstract and introduction with some information about translocations and fusions analysis in ALL

  1.     The title is ambiguous. I don’t know whether the authors aimed to emphasize that conjoined genes are common in childhood ALL or conjoined genes usually occur many times in one childhood ALL case.

Reply: We thank the reviewer for this comment and change the title as follow: ‘’Conjoined genes as common events in childhood acute lymphoblastic leukemia’’

Minor:

  1.     Typo: “Interestingly, the other 14/229 (6.1%) CGswere found to be exclusively expressed in leukemic cases but neither in solid cancers nor in normal samples …”

Reply: We corrected the typo in the text, as well as other little mistakes.

  1.    [Abstract] “Among them, we identified a subset of 14 CGs (6.1%) exclusively expressed in leukemic cases but not in solid cancers neither in normal samples.”: “neither” --> “nor”

Reply: We corrected it in the text, as well as other little mistakes.

Round 2

Reviewer 1 Report

All my points have been adequately addressed. 

Author Response

no further comments to answer

Reviewer 2 Report

The authors provided much more detailed information about the CGs, which certainly made the results clearer, but also provided new questions:

1.       The authors used dry lab methods (RNA sequencing data) to find many CGs, these CGs should be validated by wet-lab methods (such as RT-PCR, Sanger sequencing, FISH, etc.). In the manuscript, authors mentioned they used RT-PCR and/or FISH assays, and sanger sequencing wet lab method to validate the CGs, authors also provided some of RT-PCR gel electrophoresis images, but these images don't make any sense without sanger sequencing results. Moreover, Figure 2B is inconsistent with Table 3. No FISH assays are mentioned in Table 2-4 and no FISH assay results are shown in the manuscript.

2.       The column of “Validation” in Table 2-4 should delete methods (GeneBank, RefSeq, Ensembl, UCSC, etc.) which only show CGs were found in other studies and can’t be considered as a validation method to prove that the CGs exist in samples in this study.

3.       According to the supplementary Figures 1-27, there are many low-quality GCs for which most of the supporting spanning reads on the 3’ or 5’ gene side are less than 10bp, or almost all supporting spanning reads are of the same start and end positions. In this case, the authors need to use wet-lab methods to validate the CGs.

4.       There are still many typos in the manuscript, such as Line 247, Line 344, etc.
